# Returned Rate and Changed Patterns of Systemic Antibiotic Use in Ambulatory Care in Hungary after the Pandemic—A Longitudinal Ecological Study

**DOI:** 10.3390/antibiotics13090848

**Published:** 2024-09-05

**Authors:** Helga Hambalek, Mária Matuz, Roxána Ruzsa, Erika Papfalvi, Róbert Nacsa, Zsófia Engi, Márta Csatordai, Gyöngyvér Soós, Edit Hajdú, Dezső Csupor, Ria Benkő

**Affiliations:** 1Institute of Clinical Pharmacy, Faculty of Pharmacy, University of Szeged, 6725 Szeged, Hungary; ruzsa.roxana@szte.hu (R.R.); papfalvi.erika@med.u-szeged.hu (E.P.); nacsa.robert@szte.hu (R.N.); zsofia.engi@szte.hu (Z.E.); csatordai.marta@szte.hu (M.C.); soosgyongyver@szte.hu (G.S.); csupor.dezso@szte.hu (D.C.); benko.ria@med.u-szeged.hu (R.B.); 2University Pharmacy Department, Albert Szent-Györgyi Health Center, University of Szeged, 6725 Szeged, Hungary; 3Department of Internal Medicine Infectiology Unit, Albert Szent-Györgyi Health Centre, University of Szeged, Állomás Street 1-3, 6725 Szeged, Hungary; horvathne.hajdu.edit@med.u-szeged.hu; 4Institute for Translational Medicine, Medical School, University of Pécs, 7624 Pécs, Hungary; 5Emergency Department, Albert Szent-Györgyi Health Centre, University of Szeged, Semmelweis Street 6, 6725 Szeged, Hungary

**Keywords:** antibiotics, COVID-19, after the pandemic, outpatient sector, DDD, DID, antibiotic utilization, trend, pattern, Hungary, quality of use, quality indicators, AWaRe classification

## Abstract

The COVID-19 pandemic affected the epidemiology of infectious diseases and changed the operation of health care systems and health care seeking behavior. Our study aimed to analyze the utilization of systemic antibiotics in ambulatory care in Hungary after the COVID-19 pandemic and compare it to the period before COVID. We defined three periods (24 months each): Before COVID, COVID, and After COVID. Monthly trends in systemic antibiotic (J01) use were calculated using the WHO ATC-DDD index and expressed as DDD/1000 inhabitants/day (DID) and number of exposed patients per active agent. The data were further categorized by the WHO AWaRe classification. In the After COVID period, we detected almost the same (11.61 vs. 11.11 DID) mean monthly use of systemic antibiotics in ambulatory care compared to the Before COVID period. We observed a decrease in the seasonality index in the After COVID period (46.86% vs. 39.86%). In the After COVID period, the use of cephalosporins and quinolones decreased significantly, while in the case of macrolides, a significant increase was observed compared to the Before COVID period, with excessive azithromycin use (66,869 vs. 97,367 exposed patients). This study demonstrated significant changes in the pattern of ambulatory care antibiotic use in Hungary.

## 1. Introduction

Antimicrobial resistance (AMR) is one of the most significant public health problems. Its consequences are considered as devastating as climate change, the other, intertwined global challenge [1,2]. The development of AMR largely depends on the use of antibiotics, and we can consider inappropriate antibiotic use as one of the main drivers of AMR [3,4]. The COVID-19 pandemic and the consequent governmental restrictions influenced the epidemiology of other infectious diseases, changed the operation of health care systems, and affected health care seeking behavior [5,6]. During the pandemic, AMR silently increased further, and antibacterial stewardship activities were limited [7]. Several studies have addressed antibiotic utilization during the COVID pandemic, but to the best of our knowledge, very few publications have examined the antibiotic utilization pattern after the pandemic and compared it to preceding periods [8]. Hence, this study aimed to fill in this research gap and assess the trends and patterns in antibiotic utilization in the Hungarian outpatient sector before, during, and after the pandemic.

## 2. Results

During the entire study period of 72 months, 244 million DDDs of systemic antibiotics were used in the outpatient sector in Hungary. The national use of systemic antibiotics was 11.61 DDD/1000 inhabitants/day (DID) before the COVID period. During the COVID period, this decreased to 8.99 DID (22.57% decrease), and in the After COVID period, it returned to 11.11 DID.

Table 1 summarizes the monthly means of antibiotic use. The highest absolute use was observed for the penicillins group (WHO’s Anatomical Therapeutic Chemical Classification Index (ATC code: J01C)) across all three periods. However, the use of this antibiotic subgroup decreased during the COVID period, from 4.03 to 2.93 DID (27.3% decrease). After the pandemic, the level of use increased again, and even exceeded the Before COVID averages (4.12 DID). This trend was also observed for the most frequently used penicillin subgroup, the J01CR ATC subgroup. In contrast, in the use of other penicillin subgroups, we observed a continuous decrease during the study periods. In the case of beta-lactamase sensitive penicillins (ATC code: J01CE), we observed marginal use during all three periods. During the COVID period, there was a significant reduction in the use of cephalosporins (J01D), decreasing from 2.00 to 1.24 DID (38.0% decrease), and in the use of quinolones, which dropped from 1.98 to 1.36 DID (31.4% decrease). After the pandemic, a slight increase in the use of both groups was observed, but their utilization levels remained below those seen in the Before COVID period. Second-generation cephalosporins (J01DC) were the most used cephalosporins in all periods. During the pandemic, we observed a decrease in the use of the J01DC group, from 1.70 to 1.01 DID (40.59% decrease), and this scale of use stagnated in the After COVID period. However, in the use of third-generation cephalosporins (ATC code: J01DD), we observed a marked increase in the After COVID period. In the case of macrolides, lincosamides group (ATC code: J01F), we observed a significant increase in the After COVID period compared to the Before COVID period, from 2.40 to 2.89 DID.

Based on the WHO AWaRe classification, we observed a decrease in the use of Watch antibiotics during the pandemic, from 5.92 to 4.45 DID (a 24.83% decrease). However, after the COVID period, their use almost returned to Before COVID levels (5.70 DID). Regarding the percental share of Access group antibiotics (Access %) of the total systemic antibiotic use (J01), we observed a rate of ~50% in all three study periods (Table 1).

The top ten list of antibacterials is presented in Table 2. The same five active agents were at the top of the list before, during, and after the pandemic, with co-amoxiclav heading the list. However, some changes were detected: azithromycin became the second most frequently used antibacterial agent during the COVID period and maintained its position after the pandemic. Cefuroxime, which held the second position in the Before COVID period, dropped to fifth place in the list after the pandemic. Both amoxicillin and sulfamethoxazole with trimethoprim disappeared, while two new broad spectra cephalosporins appeared in the top ten list: cefixime (7th place) and cefprozil (10th place). Table 2 reveals that while the proportion of Access and Watch group agents in the top ten was equal before and during COVID (five agents each from Access and Watch groups), more Watch antibacterials were in the list after the pandemic (three agents from Access group, five agents from Watch group).

The changes in the top list were also reflected in the mean monthly number of exposed patients, summarized in Table 3. In the case of amoxicillin, clavulanic acid (co-amoxiclav), and azithromycin, we observed that the minimum mean monthly number of exposed patients doubled in the After COVID period compared to the pandemic period. However, for the other three antibacterial agents listed in the table, we noted a decrease in the number of exposed patients in the After COVID period (Appendix A).

The pattern of antibacterial use changed during the three periods. Figure 1 illustrates that the highest relative use was observed in the Beta-lactam antibacterials (ATC codes: J01C, J01D). In these groups, the combinations of penicillins, including beta-lactamase inhibitors (J01CR), had the largest relative share of use during all periods. The use of narrow-spectrum beta-lactamase sensitive penicillins (J01CE) had already been declining before the pandemic, and this downward trend continued both during and after the pandemic. In the relative use of second-generation cephalosporins (J01DC), we observed a slight decrease during all three periods. However, the relative use of third-generation cephalosporins (J01DD) increased markedly in the After COVID period. In the case of macrolides (J01FA), we observed a different trend: a continuous increase in the relative use during and after the pandemic. We noted a slight decrease in the relative use of quinolones (J01M) during the pandemic, which continued thereafter.

The seasonality indexes (J01_SV) for the three study periods are displayed in Table 4. For systemic antibacterials (J01), we observed a slight increase in the seasonality index during the pandemic (from 46.86% to 53.42%), which then decreased in the After COVID period (39.68%). Among the ATC subgroups, the most significant change was observed for the macrolides (J01FA); the seasonality index increased from 60.63% to 104.74% during the COVID period but returned to the Before COVID levels after the pandemic. In all other antibacterial subgroups, we detected a decrease in the seasonality index after the pandemic (see Table 4).

In Figure 2, we display the monthly use of systemic antibiotics (J01) during the COVID-19 period and the corresponding monthly values of the Before and After COVID periods, while in Appendix A we display the monthly use of the main antibiotic subgroups.

In the After COVID period, we observed no significant changes in the monthly systemic antibiotic use compared to the Before COVID period. In both periods, there was a winter peak, but it was slightly lower in the After COVID period.

## 3. Discussion

To our knowledge, except for the ESAC-NET annual epidemiological report and an ESAC-Net surveillance paper, this study is the first to analyze ambulatory care antibiotic use in a Central European country following the COVID pandemic [10,11]. Moreover, to the best of our knowledge, this is the first study to analyze antibiotic use including the 2 years after the pandemic. These facts significantly restrict our ability to make cross-national comparisons. During the pandemic, outpatient health care systems underwent reorganization, and multiple innovations were integrated into primary care, which persisted after the pandemic [6]. In particular, telemedicine was maintained during the After COVID period, which may have had a significant impact on the prescription of antibiotic treatments. In this study, we investigated whether antibiotic use and patterns returned to the pre-pandemic levels despite these changes.

### 3.1. Scale of Systemic Antibacterial (J01) Use

In Hungary, during the COVID period, there was a significant 22.57% decrease in the use of systemic antibiotics in ambulatory care compared to the Before COVID period. Similarly to in Hungary, a reduction in antibiotic use during the COVID was recorded in the United States (US) and Canada, and from Europe, in Portugal, Belgium, and the United Kingdom (UK) [8,12,13,14,15]. In 2020, in the first year of the COVID pandemic, the European Union (EU)/European Economic Area (EEA) experienced a significant reduction (from 18.00 to 15.00 DID between 2019 and 2020) in antibiotic consumption in the ambulatory sector. However, this decrease appeared to be temporary, as by 2022, the EU/EEA mean community (outpatient, ambulatory) consumption returned to pre-pandemic levels (an increase from 15.00 to 17.80 DID between 2021 and 2022) [10,11,16,17,18,19]. This may be explained by the fact that severe social limitations were loosened at the end of the pandemic, enabling life and infectious disease epidemiology to return to normal.

Regarding systemic antibacterial use in the community sector in Hungary, we observed that after antibacterial use decreased during the pandemic, utilization levels returned to the Before COVID levels (Before COVID: 11.61 DID–COVID: 8.99 DID–After COVID: 11.11 DID). In comparison, in the United Kingdom (UK), an increase of 0.40 DID in systemic antibiotic use was observed in the primary care sector, comparing March 2019 to March 2023. However, the use of a different study design and classification (the study used the British National Formulary to define systemic antibiotics) limits comparability [8]. Regarding antibacterial subgroups, we observed a slight increase in the absolute use of penicillins (J01C) in the After COVID period compared to the Before COVID period. In the UK, a similar trend was detected [8]. Regarding cephalosporins (J01D), the observed decline in absolute use during the COVID period persisted thereafter. In contrast, in the UK, there was a slight increase in the absolute use of this subgroup after the COVID period [8]. Again, different methodologies hinder comparison. The downward trend in the absolute use of quinolones observed in Hungary was also observed in the UK [8]. Regarding macrolide use, Hungary experienced a significant increase in use during the After COVID period. For individual countries, a heterogeneous pattern was observed. In addition to Hungary, 12 other countries reported higher consumption of macrolides in 2022 compared to 2019, whereas in the UK, a slight decrease was reported [8,10].

Analyzing systemic antibiotic use (J01) based on the WHO AWaRe classification, we observed that during the pandemic, the use of the Watch antibacterials decreased (from 5.92 to 4.45 DID). However, this reduction was not sustained in the After COVID period, as their use returned to 5.70 DID in Hungarian outpatient care.

The global use of Watch group antibiotics was on the rise, as they were prescribed for outpatients for symptoms such as fever, cough, and diarrhea [20]. Reducing the inappropriate use of Watch antibiotics is crucial for the global management of antibiotic resistance. At the same time, it is important to ensure that vulnerable populations continue to have or, where it is necessary, gain improved access to the Access group antibiotics [20,21,22].

### 3.2. Seasonal Variation

A previous Hungarian study showed significant seasonal variation in the use of systemic antibiotics in ambulatory care in Hungary, with markedly higher use during the winter months [3]. A study assessing the quality of antibiotic consumption in the community sector within the EU/EEA found that Hungary had one of the highest seasonality indexes in 2017 (J01_SV%: 51.31%). In contrast, the seasonality index was 10.56% in Denmark, 11.07% in the UK, and 12.73% in Finland [23]. In our study, we observed that the seasonality index for systemic antibacterials remained high in all three periods, with a slightly lower value in the After COVID period (Before COVID: 46.86%–COVID: 53.42%–After COVID: 39.68%). In comparison, we did not find any direct data (with seasonality index) from other countries regarding the impact of the pandemic. However, in Portugal, a nearly 40% decrease was detected in antibiotic use during the winter months of 2020, compared to the averages of the previous two years [14]. The high seasonal variation suggests that antibiotics are often prescribed for self-limiting viral respiratory tract infections (RTIs) [24].

### 3.3. Pattern of Systemic Antibacterial (J01) Use

Previous Hungarian studies have indicated that beta-lactam antibiotics (penicillins, cephalosporins) consistently played a significant role in outpatient antibiotic use in Hungary [3,25]. In our study, we observed that the beta-lactam antibacterials were still the most commonly used antibacterials. These groups represented nearly 50% of outpatient antibiotic use in all three study periods. Within the penicillins group, the use of combinations of penicillins, including beta-lactamase inhibitors (J01CR), was the most significant: this subgroup had the highest absolute use in all three study periods. From the J01CR group, amoxicillin and clavulanic acid (co-amoxiclav) led the top lists, with more than 150,000 exposed patients per month in the After COVID period (more than 1.5% of the country’s population). The dominance of co-amoxiclav is suboptimal, as it has very limited indication in the Hungarian guidelines (e.g., in acute bacterial rhinosinusitis if the first-line therapy is ineffective). In contrast, in the use of penicillins with extended spectrum (J01CA), we observed a low and constantly decreasing use, despite national resistance data and guideline suggestions. The latest national resistance data from ambulatory care samples confirmed that pneumococci, the most frequent pathogen in bacterial respiratory tract infections were sensitive to aminopenicillins in 93.2%, enabling empirical use of aminopenicillins. Consequently, amoxicillin is the first-line empiric antibiotic therapy recommended in Hungarian guidelines for acute otitis media, bacterial sinusitis, and bacterial pneumonia [26,27,28,29]. In the WHO 2023 23rd Model List of Essential Medicines, amoxicillin is also the first-line therapy for many bacterial infections [30]. The use of beta-lactamase sensitive penicillins (J01CE) became marginal in Hungary during the COVID period and continued to decline in the After COVID period. This decrease was also attributed to the limited availability of these products in Hungary. Phenoxymethylpenicillin used to be available in many products, but only one pharmaceutical company’s product remained on the market, often impacted by shortages. This marginal use is suboptimal, as phenoxymethylpenicillin is still the first-line therapy in Hungary for tonsillopharyngitis caused by *Streptococcus pyogenes*, due to national resistance surveillance data which showed 100% sensitivity of *Streptococcus pyogenes* to penicillin [29,31]. According to WHO, phenoxymethylpenicillin is the first choice therapy for community-acquired pneumonia (mild to moderate), pharyngitis, and progressive apical dental abscess, and is present on the WHO’s Essential Medicine list and guideline [30]. Regarding cephalosporins, second-generation was the most commonly used subgroup. Cefuroxim use decreased, while cefprozil appeared in the top list in the After COVID period (10th place). This may have been due to the uncertain availability (i.e., shortages) of the most commonly prescribed strength of cefuroxime (500 mg) after the pandemic. However, we detected an increase in the use of third-generation cephalosporins (J01DD), and cefixime appeared in the top 10 list. As cefixime has very limited indication in primary care (recommended as first-line agent for acute uncomplicated pyelonephritis if patient can be treated as outpatient), its increased use is worrisome [32].

During this study, the use of quinolones (J01M) increased slightly in the After COVID period compared to the pandemic years, but it remained below the Before COVID levels. A positive observation is that quinolone consumption continued to decrease in the outpatient sector in Hungary. We also observed a continuous and significant decrease in the relative use of quinolones, with the smallest relative usage occurring in the After COVID period. This is in line with the European Medicines Agency (EMA)’s recommendations from 2018 to limit quinolone use, after completing a review of severe, disabling, and potentially permanent side effects associated with quinolone prescribing [33,34,35]. As in Hungary, the sensitivity of the two common uropathogens, *Escherichia coli* and *Klebsiella pneumonia,* to fluoroquinolones was below 80%, beside safety issues, national resistance data also do not support the empirical use of fluoroquinolones [29].

The relative share of macrolide (J01FA) use has shown a significant increase since the pandemic. From the macrolide subgroup (J01FA), azithromycin emerged as the second most commonly used antibacterial agent in Hungary during the pandemic, and maintained this position thereafter. This is surprising, as in Hungarian guidelines, azithromycin is recommended as the first-line empiric therapy only for atypical pneumonia or in other infections in the case of severe penicillin allergy [28,31,36]. We presume that convenient and safe use of this agent (i.e., the once-daily dose and the short, 3-day therapy duration, and the low interaction potential) make its use favorable. Moreover, the higher incidence of pertussis cases in the country might generate azithromycin use in patients with long-term coughing [36].

The WHO 13th General Programme of Work 2019–2023 set the target that at least 60% of total systemic antibiotic consumption should be “Access” group antibiotics. In Hungary, this target was not achieved in any of the periods (Before COVID: 49.51%, COVID: 51.63%, After COVID: 50.80%) [20,21,22]. Based on the ESAC-Net 2022 report, many countries, such as Denmark, Finland, and the Netherlands, achieved this target level of relative Access antibiotic use [11].

### 3.4. Strengths and Limitations

The main strengths of this research are the longitudinal nature of the dataset and the coverage of two entire years after the pandemic. Additionally, we used a population-level dataset with excellent coverage, both in terms of inhabitants and systemic antibiotic use.

We must also acknowledge the limitations of this research. Firstly, the results may not be generalizable to other countries. Secondly, we lacked access to weekly antibiotic use data and indication-linked data, which prevented us from conducting interrupted time-series analysis and analyzing infection types. Thirdly, we could not differentiate prescriptions based on face-to-face or telemedicine consultations. Lastly, we did not have data on individual prescribers or patients, so we could not compare prescribing trends and pattern changes at the individual level.

## 4. Materials and Methods

Hungary initially declared a state of COVID pandemic emergency on 11 March 2020; hence, we defined the COVID pandemic period from March 2020 to February 2022. Then, we defined two more periods: the Before COVID period (from March 2018 to February 2020), and the After COVID period (from March 2022 to February 2024). Each period involved 24 months (see Table 5).

We conducted a population-based, longitudinal ecological study. The data were obtained from the public database of the National Health Insurance Fund (NEAK) in Hungary [37]. NEAK is the sole and mandatory insurance fund in the country, covering nearly the entire population. The NEAK database includes every reimbursed antibiotic prescription filled at community pharmacies throughout Hungary. Antibiotics are prescription-only medicines in Hungary and, with few exceptions, they are all reimbursed, meaning that the NEAK database has a ~95% drug coverage for systemic antibiotics. Package-level data of redeemed prescriptions were collected at monthly intervals. This study focused on antibiotics for systemic use, as classified by the WHO’s Anatomical Therapeutic Chemical (ATC group J01) Classification Index (version 2023) [38]. The data were further categorized by ATC antibiotic subgroups and the WHO defined AWARE classification (version 2023) [20,21,22]. The package-level utilization data were converted and expressed as defined daily doses (DDD) per 1000 inhabitants and per day (DID) [39]. In each period, we calculated the monthly means of systemic antibiotic use and compared it to the corresponding monthly means of the other two periods. We also obtained data on the number of patients exposed to different antibacterial agents. Patient-level data on antibiotic use are not available in Hungary from public resources; however, the NEAK published the number of individuals who obtained an antibiotic product at least once during a given month. Annual population data were obtained from the Hungarian Central Statistical Office [40].

To assess the quality of ambulatory care antibiotic use, we used the drug-specific quality indicators developed by the European Surveillance of Antimicrobial Consumption (ESAC). Among these indicators are the seasonal variation in systemic antibiotic use “J01_SV”. This quality indicator might reflect increased systematic antibiotic use in the “winter” quarters (October–December and January–March) compared to the “summer” quarters (July–September and April–June) within a year starting in July and ending the next calendar year in June, and expressed as a percentage: [DDD (winter quarters)/DDD (summer quarters) − 1] × 100 [9,23,41,42].

Descriptive statistics were presented as the mean ± standard deviation of the mean (SD), maximum and minimum values for continuous variables, and as the count and percentage for categorical variables. Normality was tested by visual interpretations (histogram and density plot). Continuous variables were tested via the independent *t*-test. Statistical tests were performed using R statistical software version 4.2.3 (R Foundation, Vienna, Austria) and IBM SPSS software (IBM SPSS Statistics for Windows, Version 29.0, IBM Corp., Armonk, NY, USA).

## 5. Conclusions

Compared to the Before COVID levels in this population-level analysis, we detected the same scale of antibiotic use in the After COVID period in the Hungarian ambulantory care sector. However, regarding the quality of antibiotic use, we observed many negative changes: the use of broad spectra penicillin combinations (J01CR) increased and the use of narrow spectra beta-lacatamse sensitive penicillins (J01CE) decreased, the usage of second-generation cephalosporins (J01D) decreased, while the use of third-generation ones increased. The use of macrolides (J01F) also significantly increased compared to the Before COVID period. However, we also detected favourable changes: the decreased absolute and relative use of quinolone antibacterials, and the moderate but still high seasonality index. There is an urgent need to evaluate the pandemic’s effects on both the quantity and quality of antibiotic prescribing in primary care settings in other countries, in order to tailor necessary antimicrobial stewardship interventions.

## Figures and Tables

**Figure 1 antibiotics-13-00848-f001:**
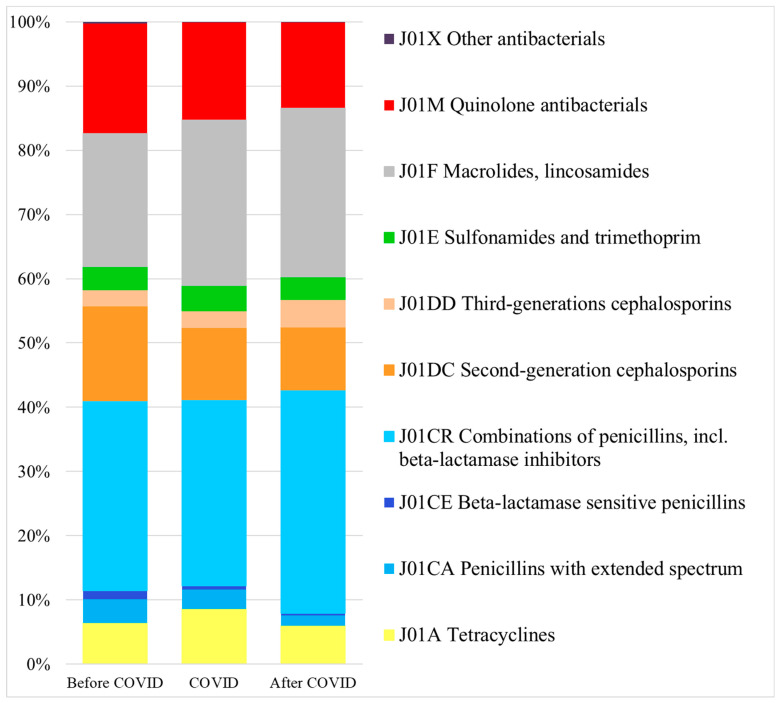
The proportional share of different antibacterial subgroups of the total systemic antibacterial use in ambulatory care during the study periods.

**Figure 2 antibiotics-13-00848-f002:**
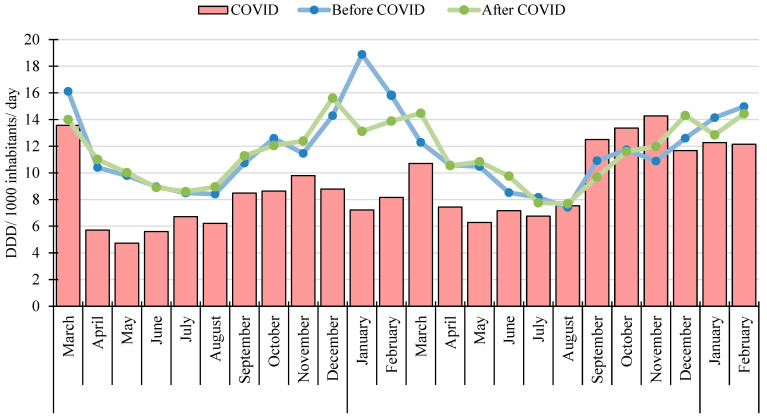
The monthly use of systemic antibacterials during the three study periods. The red columns represent the monthly use of systemic antibacterials during the COVID period, the blue lines represent the Before COVID systemic antibacterial use, and the green lines represent the After COVID systemic antibacterial use.

**Table 1 antibiotics-13-00848-t001:** The mean monthly antibiotic use in the three study periods expressed as DDD per 1000 inhabitants per day—DID).

	Before COVID Period		COVID Period		After COVID Period		*p* *
	DID (Mean) ± SD	Min–Max	DID (Mean) ± SD	Min–Max	DID (Mean) ± SD	Min–Max	
J01A Tetracyclines	0.74 ± 0.15	0.45–1.05	0.77 ± 0.22	0.46–1.17	0.67 ± 0.11	0.49–0.92	0.157
J01CA Penicillins with extended spectrum	0.43 ± 0.10	0.27–0.68	0.27 ± 0.09	0.14–0.46	0.22 ± 0.17	0.01–0.43	<0.001
J01CE Beta-lactamase sensitive penicillins	0.16 ± 0.04	0.08–0.24	0.05 ± 0.02	0.03–0.12	0.04 ± 0.03	0.04–0.03	<0.001
J01CR Combinations of penicillins, incl. beta-lactamase inhibitors	3.44 ± 0.79	2.35–5.47	2.60 ± 0.79	1.44–4.21	3.87 ± 0.84	2.68–5.47	0.024
J01C Beta-lactam antibacterials, penicillins	4.03 ± 0.94	2.70–6.39	2.93 ± 0.90	1.60–4.78	4.12 ± 0.80	3.03–5.77	0.443
J01DC Second-generation cephalosporins	1.70 ± 0.48	0.93–2.88	1.01 ± 0.33	0.53–1.86	1.05 ± 0.20	0.70–1.51	<0.001
J01DD Third-generation cephalosporins	0.29 ± 0.10	0.16–0.50	0.23 ± 0.11	0.09–0.43	0.45 ± 0.12	0.25–0.66	<0.001
J01D Cephalosporins	2.00 ± 0.57	1.10–3.39	1.24 ± 0.41	0.63–2.20	1.50 ± 0.28	1.04–2.02	0.008
J01E Sulfonamides and trimethoprim	0.42 ± 0.06	0.33–0.56	0.36 ± 0.07	0.24–0.51	0.39 ± 0.06	0.32–0.51	0.364
J01FA Macrolides	1.91 ± 0.73	0.93–3.64	1.84 ± 1.13	0.45–4.22	2.41 ± 0.83	1.09–4.06	0.007
J01FF Lincosamides	0.49 ± 0.02	0.44–0.53	0.47 ± 0.03	0.41–0.55	0.47 ± 0.02	0.44–0.53	0.004
J01F Macrolides, lincosamides	2.40 ± 0.74	1.39–4.18	2.31 ± 1.15	0.87–4.71	2.89 ± 0.84	1.54–4.46	0.009
J01M Quinolone antibacterials	1.98 ± 0.52	1.28–3.30	1.36 ± 0.34	0.86–2.06	1.52 ± 0.33	1.01–2.12	0.001
J01X Other antibacterials	0.04 ± 0.04	0.00–0.11	0.01 ± 0.00	0.00–0.01	0.01 ± 0.00	0.01–0.02	0.006
J01 Antibacterials	11.61 ± 2.90	7.41–18.88	8.99 ± 2.87	4.73–14.28	11.11 ± 2.28	7.69–15.60	0.871
J01 Access antibacterials	5.68 ± 1.14	4.09–8.54	4.53 ± 1.13	2.77–6.72	5.78 ± 0.91	4.37–7.58	0.740
J01 Watch antibacterials	5.92 ± 1.78	3.32–10.33	4.45 ± 1.78	1.95–7.83	5.70 ± 1.40	3.31–8.01	0.628
J01 Access %	49.51 ± 2.79	43.58–55.20	51.63 ± 4.68	44.38–59.62	50.80 ± 2.93	46.11–57.13	0.124

* independent sample *t* test.

**Table 2 antibiotics-13-00848-t002:** The most used top 10 antibacterials in the three periods expressed in DDD per 1000 inhabitants and per day, and as the cumulative percentage of systemic antibacterial use. The green color represents the Access group antibacterials based on the WHO’s AWARE classification, 2023.

	Before COVID Period					COVID Period					After COVID Period				
No.	ATC Code	Substance	DID ^1^	%	Cum% ^2^	ATC Code	Substance	DID ^1^	%	Cum% ^2^	ATC Code	Substance	DID	%	Cum%
1.	J01CR02	co-amoxiclav	3.43	29.55	29.55	J01CR02	co-amoxiclav	2.60	28.90	28.90	J01CR02	co-amoxiclav	4.00	34.78	34.78
2.	J01DC02	cefuroxime	1.34	11.52	41.06	J01FA10	azithromycin	1.40	15.55	44.45	J01FA10	azithromycin	1.98	17.20	51.98
3.	J01FA10	azithromycin	1.22	10.47	51.54	J01DC02	cefuroxime	0.70	8.84	53.29	J01MA12	levofloxacin	0.73	6.40	58.38
4.	J01MA12	levofloxacin	1.00	8.59	60.13	J01AA02	doxycycline	0.77	8.61	61.90	J01AA02	doxycycline	0.68	5.95	64.33
5.	J01AA02	doxycycline	0.74	6.36	66.49	J01MA12	levofloxacin	0.60	6.69	68.58	J01DC02	cefuroxime	0.64	5.58	69.61
6.	J01FA09	clarithromycin	0.66	5.65	72.13	J01FF01	clindamycin	0.47	5.27	73.85	J01FA09	clarithromycin	0.57	4.99	74.91
7.	J01MA02	ciprofloxacin	0.58	4.97	77.11	J01MA02	ciprofloxacin	0.46	5.15	79.01	J01DD08	cefixime	0.49	4.30	79.20
8.	J01FF01	clindamycin	0.49	4.25	81.36	J01FA09	clarithromycin	0.43	4.74	83.75	J01MA02	ciprofloxacin	0.48	4.20	83.40
9.	J01CA04	amoxicillin	0.43	3.70	85.06	J01EE01	SMX-TMP ^3^	0.36	3.96	87.71	J01FF01	clindamycin	0.47	4.13	87.53
10.	J01EE01	SMX-TMP ^3^	0.42	3.62	88.68	J01CA04	amoxicillin	0.27	2.99	90.70	J01DC10	cefprozil	0.45	3.90	91.93

^1^ DID: DDD per 1000 inhabitants per day; ^2^ cum%: cumulative percentage; ^3^ SMX-TMP: sulfamethoxazole and trimethoprim.

**Table 3 antibiotics-13-00848-t003:** The mean monthly number of exposed patients for the five most used antibacterial agents across the three study periods.

Mean Monthly Number of Exposed Patients					
			Before COVID Period	COVID Period	After COVID Period
J01CR02	Amoxicillin and clavulanic acid (co-amoxiclav)	mean	140,494	99,246	156,208
		min–max	89,124–227,533	51,685–166,902	97,410–220,223
J01FA10	Azithromycin	mean	66,869	66,538	97,367
		min–max	30,329–129,604	13,831–160,368	37,441–160,572
J01DC02	Cefuroxim	mean	41,890	23,251	18,391
		min–max	22,458–69,290	12,828–44,588	10,824–29,549
J01MA12	Levofloxacin	mean	39,450	22,909	28,240
		min–max	17,716–82,602	9642–44,438	13,159–46,138
J01AA02	Doxycyline	mean	8292	8185	6987
		min–max	4873–12,732	4600–13,208	4795–9914

**Table 4 antibiotics-13-00848-t004:** Seasonality index for the different antibiotic subgroups during the study periods.

Seasonality Index ^+^			
	Before COVID Period	COVID Period	After COVID Period
J01A Tetracyclines	38.92	60.29	22.48
J01CA Penicillins with extended spectrum *			
J01CE Beta-lactamase sensitive penicillins *			
J01CR Combinations of penicillins, incl. beta-lactamase inhibitors	44.52	37.97	38.53
J01C Beta-lactam antibacterials, penicillins	43.63	35.68	34.07
J01DC Second-generation cephalosporins	50.92	27.96	27.76
J01DD Third-generation cephalosporins	69.82	82.12	62.26
J01D Cephalosporins	53.39	36.26	37.32
J01E Sulfomadies and trimethoprim	28.11	30.17	22.54
J01F Macrolides, lincosamides	60.63	104.74	59.74
J01M Quinolone antibacterials	42.22	42.73	35.01
J01X Other antibacterials *			
J01 Antibacterials	46.86	53.42	39.68

* low consumption <0.5; reduced consumption; and/or a prolonged period with product shortages ^+^ Antibacterial overuse in the winter months (October–December and January–March) compared with the summer months (July–September and April–June) for a 2-year period starting in March and ending in February, expressed as percentage: [DDD (winter quarters)/DDD (summer quarters) − 1] × 100 [9].

**Table 5 antibiotics-13-00848-t005:** Visualization of the three study periods.

		Months											
		1	2	3	4	5	6	7	8	9	10	11	12
**Years**	**2018**												
	**2019**												
	**2020**												
	**2021**												
	**2022**												
	**2023**												
	**2024**												

Blue color: Before COVID period, red color: COVID pandemic period, green color: After COVID period.

## Data Availability

Data are available from the corresponding author upon reasonable request.

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
