# Peer review of "Returned Rate and Changed Patterns of Systemic Antibiotic Use in Ambulatory Care in Hungary after the Pandemic—A Longitudinal Ecological Study"

_antibiotics, 2024, doi:10.3390/antibiotics13090848_

Round 1
Reviewer 1 Report
Comments and Suggestions for Authors
The study is valuable since it uses longitudinal national antibiotic prescription data. Overall, it presents the data report clearly and demonstrates the usage trend pre-, during, and post-COVID time. Please find my specific comments below:
1. Table 1. The p-values of t-tests are confusing. The authors should explain between which data the comparisons are made. The antibiotic consumption before and after COVID?
2. It is a pity that this study does not include the antimicrobial resistance data during the same periods. I suggest the authors obtain the data in Hungary (e.g., data from the surveillance network, data from other publications) and link the usage data with the AMR data for stronger evidence and further significance of this paper.
Author Response
Response to Reviewer 1.
We thank the Reviewer for the dedicated time and for the helpful comments which enabled to improve our manuscript.
The changes are highlighted in yellow within the manuscript.
Regarding the raised questions we reply as follows:
Comment 1: Table 1. The p-values of t-tests are confusing. The authors should explain between which data the comparisons are made. The antibiotic consumption before and after COVID?
Response: Thank you for this important remark. In Table 1. the p-values of the T-test refers to the comparison of antibiotic use between the “Before COVID” and “After COVID” periods. We wrote this information below Table 1, to clarify this to the readers.
Comment 2: It is a pity that this study does not include the antimicrobial resistance data during the same periods. I suggest the authors obtain the data in Hungary (e.g., data from the surveillance network, data from other publications) and link the usage data with the AMR data for stronger evidence and further significance of this paper.
Response: Thank you for your remark. We agree that antimicrobial resistance rates and antibiotic use are interrelated and influence each other. According to your suggestion, we retrieved the national bacterial surveillance data and included resistance rates for many relevant pathogens to the discussion session of the manuscripts. Hopefully, it will satisfy you. We think that reporting further AMR rates in the result section is beyond the scope of the manuscript within the word count limits, and would not be original research (only repetition of published surveillance data). Hope that you agree with this.
We hope that these changes have improved the manuscript.
Thank you again for your work and suggestions.
Yours sincerely,
Helga Hambalek
submitting author
Reviewer 2 Report
Comments and Suggestions for Authors
Helga Hambalek et al. described changes in the use of systemic antibiotic in ambulatory care in Hungary during the pre-COVID-19, the COVID-19 pandemic, and the post-COVID-19 periods, but did not provide an in-depth analysis of the reasons for this phenomenon. A rejection is suggested for this research. Several comments are suggested as follows:
1. The introduction should detail the use of antibiotics in individual countries and highlight the importance of rational antibiotic use. The current introduction does not fulfill this task. No information on how antibiotics relate to COVID-19, or what reasons exist to connect these two things.
2 The demographic data of patients using antibiotics were not obtained in this study, and there were no clear inclusion and exclusion criteria in the Materials and Methods section. How did the authors control confounding bias? In addition, factors such as age, gender and vulnerable populations may have an impact on the use of antibiotics, but the authors did not conduct further statistical analyses. The conclusions obtained so far are not scientific.
3. Research involving human subjects should have been conducted in accordance with the World Medical Association's Declaration of Helsinki. Studies involving human participants must be performed in accordance with relevant institutional and national guidelines, with the appropriate institutional ethics committee's prior approval and informed written consent from all human subjects. No relevant material was provided in the study.
4. The authors' definitions of pre-COVID-19, COVID-19 and post-COVID-19 are unreasonable. Although Hungary initially declared a state of emergency for the COVID-19 outbreak on March 11, 2020, the COVID-19 already existed before that, so the time division of the pre-COVID-19 period is not reasonable. In addition, what evidence does the author use to determine the end time of the COVID-19 epidemic? In fact, COVID-19 remains after February 2022. The author should explain in detail the reasons for the division of time nodes.
5. The title of Table 5 resembles a figure note, lacks a proper title, and conveys limited information.
6. Figure 1: There seems to be a difference between three groups. A statistical comparison between the three groups will be interesting (Chi-squared or Fisher’s exact, as appropriate).
7. In the discussion section, the author mentioned that the use of antibiotics in other countries during COVID-19 is inconsistent with this study. Please discuss the possible reasons in detail.
8. As a longitudinal cohort study, controlling confounding bias and performing statistical analysis are the core elements. The full text fails to mention how to control confounding bias, and the data obtained are merely at the descriptive analysis stage, lacking further statistical analysis to explore the causes of this phenomenon.
Author Response
Response to Reviewer 2.
We think that reviewer 2 completely misunderstood the aim and methods of this research, hence many of the raised concerns are not relevant to this manuscript. Nevertheless, we appreciate your time and we tried to make the text more clear to avoid similar misunderstandings by other readers.
The changes are highlighted in yellow within the manuscript.
Comment 1. The introduction should detail the use of antibiotics in individual countries and highlight the importance of rational antibiotic use. The current introduction does not fulfill this task. No information on how antibiotics relate to COVID-19, or what reasons exist to connect these two things.
Response:
In the introduction we think that we clearly stated the importance of suboptimal antibiotic use in the development of AMR (which is the complementary aspect of what you suggested; line 34-41)
The use of antibiotics in other countries during and after the pandemic were clearly mentioned and referenced in the introduction (line 42-44), more detailed comparison should be done and were done in the discussion.
In our research we wished to analyze antibiotic use in relation to different time periods, with major focus on the post COVID period. We did not analyzed antibiotic use by indication including patients with COVID-19. Hence, we feel that it is not necessary to link antibiotic use and COVID in the introduction in more detail as it has been done so far (line 44-46).
Comment 2: The demographic data of patients using antibiotics were not obtained in this study, and there were no clear inclusion and exclusion criteria in the Materials and Methods section. How did the authors control confounding bias? In addition, factors such as age, gender and vulnerable populations may have an impact on the use of antibiotics, but the authors did not conduct further statistical analyses. The conclusions obtained so far are not scientific.
Response: We think that your remark is nor relevant to this research. We did not used any patient level data. We used aggregate, population level data with nearly complete (~98%) population coverage. Hence inclusion criteria, exclusion criteria and confounding bias are irrelevant.
Comment 3: Research involving human subjects should have been conducted in accordance with the World Medical Association's Declaration of Helsinki. Studies involving human participants must be performed in accordance with relevant institutional and national guidelines, with the appropriate institutional ethics committee's prior approval and informed written consent from all human subjects. No relevant material was provided in the study.
Response: Please see previous answer. Ethical approval is not needed for aggregate level drug utilization studies.
Comment 4: The authors' definitions of pre-COVID-19, COVID-19 and post-COVID-19 are unreasonable. Although Hungary initially declared a state of emergency for the COVID-19 outbreak on March 11, 2020, the COVID-19 already existed before that, so the time division of the pre-COVID-19 period is not reasonable. In addition, what evidence does the author use to determine the end time of the COVID-19 epidemic? In fact, COVID-19 remains after February 2022. The author should explain in detail the reasons for the division of time nodes.
Response: Thank you for raising this question. The COVID-19 period was defined according to the officially initiated or ended restrictions and emergency state in Hungary (which were related to the reported incidence of COVID-19 patients). According to this, we defined the COVID-19 period between March 2020 and 2022 February and extended the research 2 years before and 2 years after this period. Hope this clarify your question.
Comment 5: The title of Table 5 resembles a figure note, lacks a proper title, and conveys limited information.
Response: You are right, the title of the Table was missing. We added this in the new version of the manuscript. We believe this figure helps the reader understand the time periods and the other reviewers were ok with that, so we decided to keep it.
Comment 6: Figure 1: There seems to be a difference between three groups. A statistical comparison between the three groups will be interesting (Chi-squared or Fisher’s exact, as appropriate).
Response: We compared the groups (according to study aims: Before and After COVID) in Table 1 which contains the absolute values. We are convinced that comparison of primary data with absolute values is more proper and also more clear to understand in contrast to your suggestion (to compare relative use of antibiotic subgroups). Anyway, we believe that performing such a comparison would be a duplication, moreover the statistical methods that you suggested might not be the best to compare continuous data presented in Figure 1.
Comment 7: In the discussion section, the author mentioned that the use of antibiotics in other countries during COVID-19 is inconsistent with this study. Please discuss the possible reasons in detail.
Response: We do not understand this remark. We made clear comparisons with other countries highlighting similarities and differences with possible explanations. Please clarify to which statement you refer to and we will try to discuss it in more detail if needed.
Comment 8: As a longitudinal cohort study, controlling confounding bias and performing statistical analysis are the core elements. The full text fails to mention how to control confounding bias, and the data obtained are merely at the descriptive analysis stage, lacking further statistical analysis to explore the causes of this phenomenon.
Response: This was not a cohort study, but a longitudinal population-level study (ecological), so your remarks do not apply. We believe that we applied properly the rules of conducting drug utilization study.
Yours sincerely,
Helga Hambalek
submitting author
Reviewer 3 Report
Comments and Suggestions for Authors
Thank you very much for allowing me to review this work. It is a well-developed work on the use of antibiotics during the covid-19 pandemic, comparing it with a period before the pandemic, and another equal period, after the pandemic. The work is one of the few that allow the evaluation of the use of antibiotics at the outpatient care level, in order to have a hypothesis of what happened during the pandemic in terms of the use of the different ATC of antibiotics. I would like the authors to clarify several doubts that arise when reading the article:
At a general level, the type of work in question should be defined in the first part of the article. The terminology referring to SARS-CoV-2 infection should also be homogenized (sometimes the authors refer to it as covid; other times as covid-19)
1.- Title: the title seems more like a conclusion than a title. It should be reformulated to a title that starts with "Use of antibiotics during the covid-19 pandemic...."
2.- Abstract: It should be structured. Otherwise, it seems correct to me.
3.- Introduction: It should be developed a little more. It should be structured in 3 paragraphs, and this section should focus more on the use of antibiotics during the pandemic (there is sufficient bibliography at an international level). This would allow us to focus and better understand the theoretical framework of the study.
4.- Results: in general good. However, there is a lot of text that repeats the information in the tables. The text can be greatly reduced, or text can be used to try to explain results.
5.- Discussion: During the covid pandemic there was an erratic use of antibiotics, both at the outpatient care level and at the hospital care level. I miss that the authors have not put their results in context with the international bibliography on this topic.
6.- Methodology: From a methodological point of view, this work is an ecological study, a time series. I think it should be defined in this way.
Thank you very much
Author Response
Response to Reviewer 3.
We thank the Reviewer for the dedicated time and for the helpful comments which enabled to improve our manuscript.
The changes are highlighted in yellow within the manuscript.
Regarding the raised questions, we reply as follows:
Comment 1: At a general level, the type of work in question should be defined in the first part of the article. The terminology referring to SARS-CoV-2 infection should also be homogenized (sometimes the authors refer to it as covid; other times as covid-19)
Response: Thank you for your remark. We defined study type in the subtitle of the manuscript. We harmonized the name of the COVID infection throughout the manuscript text to be consequent.
Comment 2: Title: the title seems more like a conclusion than a title. It should be reformulated to a title that starts with "Use of antibiotics during the covid-19 pandemic...."
Response: Thank you for your remark. We think that this question is a matter of taste. We dared to choose to use this title because an ECDC manuscript (DOI: 10.2807/1560-7917.ES.2023.28.46.2300604) also employed a similar style.
Comment 3: Abstract: It should be structured. Otherwise, it seems correct to me.
Response: Thank you very much. We followed the author instruction guideline of the journal that requires unstructured abstract (We also favour structured abstracts).
Comment 4: Introduction: It should be developed a little more. It should be structured in 3 paragraphs, and this section should focus more on the use of antibiotics during the pandemic (there is sufficient bibliography at an international level). This would allow us to focus and better understand the theoretical framework of the study.
Response: We would like to focus our research on the post COVID period and whether pre COVID antibiotic use return in scale and pattern after the pandemic. That is why we did not pay more attention for the COVID period (moreover this has been already done in a previous manuscript: DOI: 10.3390/antibiotics12060970). If you clarify the three sections you would like to read, we are happy to complement the current version of the introduction.
Comment 4: Results: in general good. However, there is a lot of text that repeats the information in the tables. The text can be greatly reduced, or text can be used to try to explain results.
Response: You are completely right about this. We tried to make the result section more concise and we deleted many numbers which were repetition of the values in the table.
Comment 5: Discussion: During the covid pandemic there was an erratic use of antibiotics, both at the outpatient care level and at the hospital care level. I miss that the authors have not put their results in context with the international bibliography on this topic.
Response: You are right, comparison of antibiotic use during the pandemic is an extremely important topic. However, as we have already done it in a previous manuscript (doi:10.3390/antibiotics12060970.) so we tried to avoid any overlap and we tried to focus on the post COVID period (where comparison possibilities were limited due to the lack of published data from many countries as we mentioned).
Comment 6: Methodology: From a methodological point of view, this work is an ecological study, a time series. I think it should be defined in this way.
Response: Thank you for this remark. We defined study design accordingly in first part of the methods section and in the manuscript subtitle.
We hope that these changes have improved the manuscript.
Thank you again for your work and suggestions.
Yours sincerely,
Helga Hambalek
submitting author
Round 2
Reviewer 3 Report
Comments and Suggestions for Authors
Thank you very much for clarifying my doubts. I now believe that the work meets the conditions for publication.